# POSS Compounds as Modifiers for Rigid Polyurethane Foams (Composites)

**DOI:** 10.3390/polym11071092

**Published:** 2019-06-27

**Authors:** Anna Strąkowska, Sylwia Członka, Krzysztof Strzelec

**Affiliations:** Institute of Polymer & Dye Technology, Lodz University of Technology, 90-924 Lodz, Poland

**Keywords:** RPUF, POSS, reinforcing effect, thermal properties, morphology

## Abstract

Three types of polyhedral oligomeric silsesquioxanes (POSSs) with different functional active groups were used to modify rigid polyurethane foams (RPUFs). Aminopropylisobutyl-POSS (AP-POSS), trisilanoisobutyl-POSS (TS-POSS) and octa(3-hydroxy-3-methylbutyldimethylsiloxy-POSS (OH-POSS) were added in an amount of 0.5 wt.% of the polyol weight. The characteristics of fillers including the size of particles, evaluation of the dispersion of particles and their effect on the viscosity of the polyol premixes were performed. Next, the obtained foams were evaluated by their processing parameters, morphology (Scanning Electron Microscopy analysis, SEM), mechanical properties (compressive test, three-point bending test, impact strength), viscoelastic behavior (Dynamic Mechanical Analysis, DMA), thermal properties (Thermogravimetric Analysis, TGA, thermal conductivity) and application properties (contact angle, water absorption). The results showed that the morphology of modified foams is significantly affected by the fillers typology, which resulted in inhomogeneous, irregular, large cell shapes and further affected the physical and mechanical properties of the resulting materials. RPUFs modified with AP-POSS represent better mechanical properties compared to the RPUFs modified with other POSS.

## 1. Introduction

Polyurethanes (PUs) are a group of polymers characterized by the most diverse properties, thanks to which they have a very wide range of industrial applications [1,2,3]. For the synthesis of polyurethanes, isocyanates that were obtained by Wurtz in 1849 are used as the fundamental raw material [4]. Among the synthesized polymeric materials, PUs currently occupy fifth place among the most commonly used plastics in the world and constitute 7.7% of total plastics produced [5,6]. The global PU market is dominated by PU foams, which account for over 65% of global production. Worldwide production of PU foams mainly includes elastic foams, which cover about 37% of world production, making them the most comprehensive group of polyurethane and rigid materials, which immediately after flexible foams, are the second largest group of polyurethane plastics and constitute about 28% of global polyurethane production [7]. Rigid polyurethane foams (RPUFs) are used as high-performance thermal insulation materials in construction, pre-insulated pipelines and in the refrigeration industry due to their properties, such as closed cell structure, low thermal conductivity, and low moisture absorption capacity [8,9,10,11]. Open-cell polyurethane foams increase the heat transfer capacity, which means they are characterized by much higher thermal conductivity than RPUFs with a closed-cell structure [12]. The thermal insulation properties of RPUFs also depend on the apparent density of the foam, which has a fundamental influence on the heat conduction coefficient. The values of apparent density for RPUFs with an open cell structure is 10 to 12 kg m^−3^, while for closed cell foams the value usually lies in the range 25–70 kg m^−3^. The smallest values of thermal conductivity are observed for RPUFs with closed-cell structure, the apparent density of which is from 25 to 35 kg m^−3^ [13]. A significant impact on the wide range of applications of rigid PUR foams also has mechanical and physical properties. However, PU materials are flammable, giving off toxic gases during combustion that pose a threat to the surrounding environment. Fire resistance should be limited so that it can be successfully used in building applications [14,15,16,17,18].

A review of the related literature indicates that one of the method of increased flammability of PU foams are fire-resistant coatings. For example, a layer-by layer assembly technique of polyethylenimine (PEI), graphene oxide (GO) and synthetic melanin nanoparticles (SMNPs) was constructed on the surface of PU foams [19]. Obtained materials exhibited an excellent radical scavenging capacity and high-temperature stability. Cho et al. employed polydopamine into the flame-retardant surface coating system of the flexible PU foam, realizing a decreased peak heat release rate by 67% [20]. Roberts et al. [21] further analyzed the thermal degradation kinetics of polydopamine-nanocoated PU foam and discovered that polydopamine could consistently decrease the release amount of combustible volatiles [22]. On the other hand, another studies have shown that better improvement of mechanical properties and heat resistance, can be also obtained by use of nanomaterials combined with PU matrix [18,22,23,24,25,26,27,28].

It was reported that, in addition to traditional fillers, an interesting group of reactive nanofillers are polyhedral oligomeric silsesquioxanes (POSS), which combine the features of organic and inorganic materials [29,30,31]. They were discovered at the beginning of the 20th century and are currently in the interest of many research centers due to their unique properties, including well-defined, three-dimensional, chemically and thermally resistant hybrid structure with nanoscopic dimensions [32,33,34,35]. Their biggest advantage is the ease of functioning with various organic or inorganic substituents. The most frequently occurring substituents are hydrogen, amine, acrylic and methacrylic, vinyl, allyl, epoxide, hydroxyl or halogen groups [36,37]. Thanks to the hybrid structure and the diversity of function groups, POSS exhibits comprehensive chemical compatibility [38,39]. POSS has a regular polyhedral spatial structure presented as a regular cube. The size of POSS depends on their construction. The size of the octasilsesquioxane core is about 0.5 nm, and the whole particle reaches dimensions of 1 to 3 nm depending on the type of substituents [40]. The oligosilsesquioxanes in the polymer matrix tend to agglomerate so that crystals can form in the polymer matrix [41]. Their dimensions can range from a dozen to several dozen nanometers [42,43]. Due to their properties, POSS compounds are commonly used in multifunctional applications. In contrast to the traditional organic compounds, they do not emit volatile organic substances, thanks to which they are odorless and environmentally friendly [40]. The basic element deciding about the potential application and influencing the final properties of the material are the organofunctional substituents found in the corners of the silicon-oxygen skeleton, and more specifically their number, distribution, and chemical nature. Thanks to the combination of reactive and non-reactive substituents POSS molecules can be easily attached to the polymer by copolymerization, transplantation or simply mixing [44].

Hybrid nature and nanometric size scale make silsesquioxanes extremely attractive materials used as nanofillers [45]. It was shown in previous studies that the introduction of POSS compounds to the polymeric material may increase the temperature of use and decomposition, affect the glass transition temperature, increase the resistance to oxidation of the composite, improve mechanical strength, affect surface hardening, reduce flammability and heat [43,46,47].

The aim of this research is to develop and test PU foams modified with three type of POSS—two of them with closed-cage functionalized by amino- or hydroxyl group (Aminopropylisobutyl-POSS (AP-POSS) and octa(3-hydroxy-3-methylbutyldimethylsiloxy-POSS (OH-POSS) and one with open-cage (trisilanoisobutyl-POSS (TS-POSS)). The influence of the different type of POSSs on thermal properties (Thermogravimetric Analysis, TGA), dynamic mechanical properties (Dynamic Mechanical Analysis, DMA), physico-mechanical properties (compression strength, three-point bending test, impact strength, apparent density, water absorption) and morphology of obtained PU composites was examined in current work. The obtained results, indicate that the addition of POSS influences the morphology of analyzed foams and consequently their further mechanical and thermal properties.

## 2. Experimental Section

### 2.1. Materials

The water-blown PUR foams used in this study were obtained from a two-component system supplied by Purinova Sp. z o. o. from Bydgoszcz, Poland, after mixing the polyol (Izopianol 30/10/C) and the diphenylmethane diisocyanate (Purocyn B). The polyol is a mixture of components containing polyester polyol (hydroxyl number ca. 230–250 mgKOH/g, functionality of 2), catalyst (*N*,*N*-Dimethylcyclohexylamine), flame retardant (Tris(2-chloro-1-methylethyl)phosphate), chain extender (1,2-propanediol) and water as blowing agent [48]. This formulation was selected because of its industrial relevance for the production of PU insulation ensures that the raw materials are available from a range of manufacturers at a competitive cost. AP-POSS, TS-POSS and OH-POSS were provided by Hybrid Plastics Inc. (Hattiesburg, MS, USA). The chemical structures of AP-POSS, TS-POSS and OH-POSS are presented in Figure 1a,b,c respectively.

### 2.2. Manufacturing of RPUFs

RPUFs containing 0.5 wt.% of AP-POSS, TS-POSS, and OH-POSS were obtained as follows. The adequate amount of selected POSS was added to the Izopianol 30/10/C and the mixture (component A) was homogenized with an overhead stirrer at 600 RPM under ambient conditions for approximately 60 s. Purocyn B (component B) was added to component A and the mixture was stirred for 10 s at a speed rate of 1800 RPM. Following the information provided by the supplier, the ingredients were mixed in the ratio of 100:160 (ratio of component A to component B). Such prepared system was poured into an open mold and allowed to expand freely in the vertical direction. RPUFs were conditioned at room temperature for 24 h. After this time, samples were cut with a band saw into appropriate shapes (determined by obligatory standards listed below in the *Characterization techniques*) and their physico-mechanical properties were investigated. A schematic figure of the synthesis of RPUFs is presented in Figure 2.

### 2.3. Characterization Techniques

The average size of POSS powder particles was measured using a Zetasizer NanoS90 instrument (Malvern Instruments Ltd., UK). The size of particles polyol dispersion (0.04 g/L) was determined with the dynamic light scattering DLS method.

The absolute viscosities of polyol and isocyanate were determined corresponding to ASTM D2930 (equivalent to ISO 2555) using a rotary Viscometer DVII+ (Brookfield, Germany). The torque of samples was measured as a range of shear rate from 0.5 to 100 s^−1^ at room temperature.

The apparent density of foams was determined accordingly to ASTM D1622 (equivalent to ISO 845). The densities of five specimens per sample were measured and averaged.

The morphology and cell size distribution of foams were examined from the cellular structure images of foam which were taken using JEOL JSM-5500 LV scanning electron microscopy (JEOL Ltd., USA). All microscopic observations were made in the high-vacuum mode and at the accelerating voltage of 10 kV. The samples were scanned in the free-rising direction. The average pore diameters, walls thickness, and pore size distribution were calculated using ImageJ software (Media Cybernetics Inc.).

The thermal properties of the synthesized composites were evaluated by TGA measurements performed using a STA 449 F1 Jupiter Analyzer (Netzsch Group, Germany). About 10 mg of the sample was placed in the TG pan and heated in an argon atmosphere at a rate of 10 K min^−1^ up to 600 °C with the sample mass about 10 mg. The decomposition temperatures (*T_5%_*, *T_10%_*, *T_50%_* and *T_70%_* of mass loss) were determined.

The compressive strength (*σ_10%_*) of foams was determined accordingly to the ASTM D1621 (equivalent to ISO 844) using Zwick Z100 Testing Machine (Zwick/Roell Group, Germany) with a load cell of 2 kN and the speed of 2 mm min^−1^. Samples of the specified sizes were cut with a band saw in a direction perpendicular to the foam growth direction. Then, the analyzed sample was placed between two plates and the compression strength was measured as a ratio of the load causing 10% deformation of sample cross-section in the parallel and perpendicular direction to the square surface. The result was averaged of 5 measurements per each sample.

Impact test was carried out in agreement with ASTM D4812 on the pendulum 0.4 kg hammer impact velocity at 2.9 m s^−1^ with the sample dimension of 10 × 10 × 100 mm. All tests were performed at room temperature. At least five samples were prepared for the tests.

Three-point bending test was carried out using Zwick Z100 Testing Machine (Zwick/Roell Group, Germany) at room temperature, according to ASTM D7264 (equivalent to ISO 178). The tested samples were bent with testing speed 2 mm min^−1^. Obtained flexural stress at break (*ε_f_*) results for each sample were expressed as a mean value. The average of 5 measurements per each type of composition was accepted.

Dynamic mechanical analysis (DMA) was determined using ARES Rheometer (TA Instruments, USA). Torsion geometry was used with samples of thickness 2 mm. Measurements were examined in the temperature range 20–250 °C at a heating rate of 10 °C min^−1^, using a frequency of 1 Hz and applied deformation at 0.1%.

Surface hydrophobicity was analyzed by contact angle measurements using the sessile-drop method with a manual contact angle goniometer with an optical system OS-45D (Oscar, Taiwan) to capture the profile of a pure liquid on a solid substrate. A water drop of 1 μL was deposited onto the surface using a micrometer syringe fitted with a stainless steel needle. The contact angles reported are the average of at least ten tests on the same sample.

Water absorption of the RPUFs was measured according to ASTM D2842 (equivalent to ISO 2896). Samples were dried for 1 h at 80 °C and then weighed. The samples were immersed in distilled water to a depth of 1 cm for 24 h. Afterward, the samples were removed from the water, held vertically for 10 s, the pendant drop was removed and then blotted between dry filter paper (Fisher Scientific, USA) at 10 s and weighed again. The average of 5 specimens was used.

Changes in the linear dimensions were determined in accordance to the ASTM D2126 (equivalent to ISO 2796). The samples were conditioned at the temperature of 70 °C and −20 °C for 14 days. Change in linear dimensions was calculated in % from Equation (1).
Δ*l* = ((*l* − *l_o_*)/*l_o_*) × 100(1)
where *l_o_* is the length of the sample before thermostating and *l* is the length of the sample after thermostating. The average of 5 measurements per each type of composition was reported.

## 3. Results and Discussion

### 3.1. Average Size of POSS Powder Particles and the Dispersion of POSS-Modified Polyol Premixes

One of the most important parameters determining the behavior of the filler in the polymer matrix is the size of its particles. If the particles are too small, their dispersion may be difficult because they have a greater tendency to aggregate and agglomerate, forming large clusters in the matrix. Too large particles may affect the foaming process and further properties of the obtained materials. The particle size of the POSS powder was measured in a polyol dispersion (0.04 g/L). The results of particle size measurements are given in Figure 3.

From the diagram, it follows that the size of AP-POSS particles ranges from 65 to 104 nm, the highest percentage—29% is shown by 82 nm particles. In the case of TS-POSS, the particle size distribution is somewhat larger and ranges from 59 to 108 nm, with the largest 69 nm volume fraction. Such small particle sizes of nanofillers may suggest their tendency to agglomerate in the polyol, which may negatively affect mechanical and functional properties. Figure 4 shows the optical micrographs obtained for the polyol systems with AP-POSS, TS-POSS and additionally OH-POSS. A comparison of the optical images for the sample with AP-POSS (Figure 4a) with that of TS-POSS (Figure 4b) reveals that in both cases the particles are well dispersed in the polyol systems and no aggregates of the POSS’s particle are observed. A different trend is observed for the sample with OH-POSS. As presented in Figure 4c, a homogenous dispersion of the polyol system is observed, as a result of the liquid character of the used OH-POSS.

### 3.2. Impact of POSS on PU Mixture Viscosity

The viscosity of the reactive mixture was measured first since it is a critical parameter affecting the foaming process [49]. Increased viscosity hinders bubble growth, yielding foams with lower cell size. Table 1 presents the results of the change in dynamic viscosity depending on the type of POSS in polyol mixture. The polyol premixes that contained AP-POSS, TS-POSS and OH-POSS are characterized by an increase in their viscosity, as a result of the presence of POSS particles interacting with the polyether polyol through hydrogen bonding and van der Wall’s interaction [48]. Compared to control polyol, the greatest dynamic viscosity has AP-POSS modified polyol mixture.

The rheological properties of polyol premixes are shown as the viscosity versus shear rate in Figure 5a. In all systems, the viscosity is generally reduced at increased shear rates. Such a phenomenon is typical for non-Newtonian fluids with a pseudoplastic nature and is quite often found in the many previous works [50,51]. To further analyses the data, graph of viscosity versus shear rate is converted to log viscosity versus log shear rate form as shown in Figure 5b. It can be seen that the curvatures of viscosity versus shear rate can be made close to linear using this log-log format with regression of 0.979–0.982. The power law index (*n*) was calculated from the slopes. All results are presented in Table 1. For the system containing AP-POSS, the power law index is lower than that of their TS-POSS and OH-POSS modified system counterparts. It indicates that the effect of the filler on the pseudoplasticity behavior becomes more significant for systems modified with AP-POSS, leading to the highly non-Newtonian behavior.

### 3.3. The Influence of POSS on the Maximum Temperature (T_max_) of the Reaction Mixture during the Foaming Process

The reaction of the synthesis of RPUFs is highly exothermic [15,52]. The rate of increase in temperature determines the activity of reaction mixture that is associated with the reactivity of the components of the mixture. As shown in Table 2, the introduction of AP-POSS, TS-POSS, and OH-POSS into the PU system increases the activity of reaction mixture which is confirmed by an increase in the *T_max_* during the foaming process in each case. The presence of the additional groups as a result of the incorporation of the filler can lead to the exothermic reaction providing more heat evaporated to the system, and consequently higher temperature of the modified system compared to the PU-0. The *T_max_* increases by about 20 °C with the addition of each POSS and appears at longer times compared to the PU-0 (Figure 6). Basically, an analog tendency was observed by other authors in previous works [16,53,54].

### 3.4. Foaming Kinetic of RPUFs

The foaming process was determined by measuring the characteristic processing times like cream, extension and gelation time. The cream time was measured from the start of mixing of components to a visible start of foam growth, extension time elapsing until reaching the highest volume of the foam and gelation time was determined as the time when the foam solidifies completely and the surface is no longer tacky [17]. The results presented in Table 2 indicate a slight increase in cream and extension time for the RPUFs containing AP-POSS, TS-POSS, and OH-POSS. This dependence is mostly related to the fact that well-dispersed filler in the reaction mixture acts as a nucleating agent and higher viscosity of the modified systems is observed. It was reported in a previous work that higher viscosity has a major impact on the growth of RPUFs and causes an increase in reaction time by a few minutes [39]. Also, an increase of filler content affects the kinetics of the reaction and the phase separation. The rate of PU polymerization during foaming and morphology development is slowed down [40]. The addition of the filler into the system decreases the rate of isocyanate conversion during the early stage reaction. Also, due to the presence of the filler, a reduction of the mobility of the molecules takes place [40], leading to prolonged cream and extension time [18,22]. Compared to the PU-0, composites modified with the addition of the fillers are also characterized by a shorter tack-free time, indicating that filler particles act as a curing accelerator. Among studied fillers, the highest values of extension time and tack-free time are determined for PU-AP composites, as a result of higher viscosity, as compared to the PU-TS and PU-OH counterparts.

### 3.5. Density of RPUFs

Apparent density is one of the most important parameters to control the physical, mechanical and thermal properties of the RPUFs which has influence on their performance and applications. The values of density of prepared foams are presented in Table 2. In general, term, the apparent density tends to increase when the POSS are added. PU-0 is characterized by an apparent density of 38 kg m^−3^. The apparent density increases to 43, 42 and 40 kg m^−3^ for samples with AP-POSS, TS-POSS, and OH-POSS, respectively. This effect can be explained by an analysis of the role of filler particles on nucleation and cell growth. The POSS particles act as nucleation sites promoting the formation of bubbles, and this is an increasing trend with nanoparticles content, but, at the same time, the growth process of the resulting cells is hindered by an increase of the gelling reaction speed, revealing in bigger viscosity. This results in bubble collapse and higher density foams. Moreover, the reactive groups of POSS particles (such as, hydroxyl and amine groups) would react with isocyanate (–NCO) groups, therefore, the content of isocyanates which reacted with water and produced CO_2_ foaming gas would decrease, leading to the decreased foaming ratio and increased density. To sum up, the density of the composite foams increased with the incorporation of POSS filler in the PU system.

### 3.6. Morphology of RPUFs

The cell morphology is one of the most important factors determining the physico-mechanical properties of RPUFs [23,51]. The foaming process, the formation of cells, and their shape can be explained by a nucleation and growth mechanism [24]. A proper balance of filler concentration, reaction temperature, viscosity and dispersion of the filler in the polymer matrix is the key for optimization of the cellular structure of RPUFs [25]. The cellular structures of RPUFs composites are presented in Figure 7.

As observed from the micrograph of the neat PU-0 (Figure 7a,b), the cell size and cell distribution are nearly uniform and the PU-0 consists of closed cells with a negligible amount of cells with broken walls. With the addition of AP-POSS, the overall cell structure becomes less uniform and the number of broken cells increased (Figure 7c,d). A similar trend is observed for sample PU-TS, as shown in Figure 7e,f, although it has a higher content of broken cells compared to PU-AP. The more homogenous structure is observed in Figure 7g,h, which corresponds to the PU-OH. The closed-cell structure is well-preserved, and the number of broken cells is decreased. Higher content of open cells in the case of RPUFs modified with AP-POSS and TS-POSS can be connected with poor interfacial adhesion between the filler surface and the polymer matrix, which promotes earlier cell collapsing phenomena and increases a high possibility of generating open pores [26]. Moreover, the possible interphase interactions between POSS and PU in cell struts disturbed formulation of stable foam structure [27] which results in the coalescence of crowded cells. The alteration of cell morphology as the result of filler incorporation was also observed in previous studies [28,35,55].

The values of the cell size of the RPUFs were statistically analyzed by means of *ImageJ* software from SEM images and the median values are summarized in Table 2. The cell size distribution of the RPUFs is presented in Figure 8. From the table, the PU-0 has fewer cells with a larger cell size than the POSS-modified composites. In general, the PU-0 has an average cell size of 466 µm, and the addition of small amounts of POSS yielded smaller cells. RPUFs with AP-POSS, TS-POSS, and OH-POSS have a cell size of 396 µm, 389 µm and 408 µm, respectively. This means that RPUFs composites containing POSS have a higher cell density and smaller cell size than those of the PU-0. Therefore, it can be concluded that the POSS addition has an effect on reducing the cell size. This may be due to the increased viscosity of the system after POSS addition which restrains the expansion of the cells. Moreover, it has been well established in previous works that filler particles can act as nucleation sites for cell formation and since a higher number of cells starts to nucleate at the same time, thus a higher amount of cells with reduced diameter is present [56,57,58,59,60,61].

### 3.7. Compressive Strength of RPUFs

The mechanical properties of RPUFs depend primarily on the cells’ morphology with the strength being higher in the direction of foam expansion. In Figure 9 it can be seen that all the compressive stress-strain plots of RPUFs are composed of a first linear region which corresponds to the elastic response of the material and a second region in which the curves present a large plateau due to the plastic deformation and rupture of the cell walls while the stress is constant until the cells are crushed. Nevertheless, some differences can be observed between the samples. The increase in brittleness caused by the reinforcements determines a more abrupt transition from the elastic region to the plateau, in contrast to the smooth transition observed in the case of the PU-0. The elongation at break of the PU composites decreases with POSS incorporation, implying that POSS particles make the PU matrix more rigid. This is a common result in PU composites reinforced by a conventional filler [16,62,63].

The compression modulus and compressive strength of RPUFs are presented in Table 3. The compressive strength of all materials tested in the direction parallel and perpendicular to the direction of foam rise is greater than the strength of the reference foam. The largest increase in compressive strength is observed for the PU-AP and it is about 351 kPa in a parallel direction and 159 kPa in the perpendicular direction. In the foams containing TS-POSS and OH-POSS, there is a slight decrease in compressive strength compared with RPUFs containing AP-POSS; however, it is still larger than for the PU-0. As presented in Figure 10 the mechanical properties are closely related to the apparent density of polymer composites. An increase in density is accompanied by an increase in the mechanical properties of the composites since in compression the stiffness arises from buckling of cell walls. The higher density is related to more compact cellular structures, hence there is more material per unit area and the modulus and strength increase [64]. POSS-modified foams obtained in this study show apparent density values of 40–43 kg m^−3^ and compressive strengths of 309–351 kPa, which are well in the range exhibited by conventional commercial foams that present densities in the 15–130 kg m^−3^ range and compressive strength values in the range 200–220 kPa (for RPUFs at a density of 40 kg m^−3^) [60,65]. Based on these results, the foams modified with POSS can potentially be used on an industrial scale in the construction and packaging industries.

### 3.8. Flexural Strength of RPUFs

As in the case of compression results presented in Figure 10, the correlation between flexural strength (*σ_f_*) and apparent density is observed as well (Figure 11). It can be also seen that incorporation of POSS filler affects the *σ_f_* of POSS-modified materials. Compared to the PU-0, *σ_f_* is improved by the addition POSS in all cases. The value of tensile strength of PU-AP increases by 38% from 0.402 to 0.469 MPa as compared to the PU-0. Similar trend id observed for RPUFs modified with TS-POSS and OH-POSS. The value of *σ_f_* increases to 0.430 and 0.427 MPa for samples PU-TS and PU-OH. Figure 12 shows the stress-elongation curves for the RPUFs. All samples exhibit a linear elastic behavior in the low-stress region and plastic deformation in the high-stress region, pointing at the comparable mechanical performance of modified foams. The incorporation of POSS reduces the elongation at break (*ε_f_*) of RPUFs in all cases. The reason is due to the presence of POSS aggregates within the PU matrix, which may act as defects during the tensile testing process and decrease *ε_f_* of foam composites.

### 3.9. Impact Strength of RPUFs

The correlation between impact strength and apparent density is observed as well (Figure 13). With the incorporation of AP-POSS, TS-POSS, and OH-POSS, the impact strength increases from 0.35 to 0.46, 0.45 and 0.42 kJ m^−2^, respectively. This behavior is related to the good interface reinforcement matrix and the generation of fracture paths through the POSS-reinforced RPUFs. Thus, the deformability of the RPUFs matrix is reduced, which in turn affects the ductility in the foam surface. With this effect, the foam composite tends to form a more rigid structure and decrease the concentration of POSS, thus reducing the foam’s energy absorption, resulting in greater impact strength.

### 3.10. Dynamic Mechanical Analysis (DMA) and Thermogravimetric Analysis (TGA)

The dynamic mechanical behavior of RPUFs as a function of the temperature is shown in Figure 14. The results presented in Figure 14a and Table 4, indicate that the incorporation of POSS to the PU matrix affects the value of *T_g_*, which corresponds to the maximum value of the curve loss tangent (*tanδ*) versus temperature. Compared to the RPUFs modified with TS-POSS and OH-POSS, RPUFs containing AP-POSS are characterized by higher *T_g_*. Wu et al. [66] have shown that the *T_g_* of RPUFs reflects the rigidity of the polymer matrix which is a function of the isocyanate index, cross-link density and aromaticity level of the RPUFs. Given that the isocyanate index has been held constant in this study, the increase in the *T_g_* for POSS-modified samples must be a reflection of the increased aromaticity and cross-link density due to the presence of the POSS [67]. Moreover, as shown in Figure 14a, the reference and POSS-modified foams exhibit one wide peak in the range of temperature analyzed. The width of the peak becomes broader with the POSS incorporation due to different relaxation mechanisms appearing in the modified materials as a consequence of the added filler. The broadening of the *tanδ* peak is often assumed to be due to broader distribution in molecular weight between crosslinking points or heterogeneities in the network structures [19].

In Figure 14b, it is also notable that RPUFs modified with POSS are characterized by higher storage modulus (*E’*) as compared to PU-0. It can be concluded that the addition of all POSS has significantly increased the *E’* of PU and consequently the stiffness of studied composites is also enhanced. This is due to the presence of filler in the PU matrix as well as higher viscosity of the modified systems, which imposes serious limits on the mobility of polymer chains, affecting their higher stiffness. Similar results are reported in the literature [68,69].

The thermal degradation of pure polyurethane foam and hybrid composites was monitored by TGA thermograms as displayed in Figure 15a. The thermo-oxidative decomposition temperatures for 5, 10, 50 and 70% weight loss are evaluated from TGA curves, as listed in Table 4. In the case of PUR foams, thermal degradation occurred in 3 stages. In the first stage of decomposition at about 10% loss of initial mass, dissociation of urethane bonds occurs at a temperature of 150 to 330 °C [70,71]. The second degradation step RPUF corresponding to a weight loss of about 50% occurs at a temperature between 330 and 400 °C and is attributed to the decomposition of the soft polyol segments [70,72]. Then, the third degradation step associated with the degradation of the fragments generated during the second stage occurs at 500 °C, which corresponds to 80% loss of mass [70,72].

It can be observed that the addition of fillers affects the thermal stability of RPUF (Table 3). POSS used as foam modifiers is characterized by higher thermal stability and percentage losses of masses at much higher temperatures than PU-O. However, in the presence of POSS, the acceleration of mass loss at the initial stage of degradation was observed.

The reduction of thermal stability can be attributed to non-homogeneous dispersion of POSS and changes in cross-link density [51]. Confirmation is SEM photos, which clearly show that the presence of POSS increases the heterogeneity of the RPUF morphology. In further stages of degradation, modified foams are slightly more stable and are characterized by weight losses obtained at a similar temperature as pure foam. In further degradation steps, the modified foams are slightly more stable and are characterized by mass losses obtained at a similar temperature as pure foam, with maximum mass losses of approximately 314–322 °C and 551–584 °C, which is related to the reaction of oxygen with hydroperoxides that are themselves unstable and decay, creating more free radicals [73]. In addition, it can be seen that the amount of char residue for POSS filled foams is increased compared to PU-0. This results in more stable char layers that can protect materials from further decomposition and, in turn, increase thermal stability. The change is also visible on the DTG curves, which are the first derivative of TGA that represent the speed of the composite decomposition process during heating. It can be seen in Figure 15b that the degradation rate of POSS modified foams is slightly lower than that of PU-0 foam.

### 3.11. Dimensional Stability, Contact Angle and Water Absorption

For RPUFs, often used as construction materials, dimensional stability, as well as affinity for water, is a very important parameter. Table 5 and Figure 16 show the dimensional stability of foams at low (−20°C) and high temperature (70 °C), respectively. The variability of dimensions at low temperature was slightly higher than at high temperature for the same foam samples. Furthermore, the % linear changes in length, width, and thickness after exposure indicate that the addition of POSS generally resulted in smaller dimensional changes of the modified foams compared to the reference foam, indicating a stabilizing effect of POSS on the degradation factor. This is particularly evident in the conditions of elevated temperature, where for TS-POSS modified foams, the dimensional stability improved by an average of 20% in comparison with the PU-0. The only exception to this trend is the POSS-OH-modified sample, which shows slightly larger changes in linear dimensions compared to the reference sample, especially at reduced temperatures. However, according to industrial standard, PU panels tested at 70 °C should have less than 3% of linear change. In each case, the dimensional stability of PU foams is thus still considered to be mild and within commercially acceptable limits [74].

Polyhedral oligomeric silsesquioxanes significantly affected the hydrophobicity of the foams (Figure 17 and Figure 18). Regarding water absorption, it is notable that foams modified by POSS absorb less water than the reference sample. This effect is attributed to the greater surface roughness of foams with smaller pore sizes as well as the lack of large surface pores in which water droplets can be stored. Lower water absorption indicates greater hydrophobicity, which is also well illustrated by the contact angles of foam surfaces with water (Figure 18). The most hydrophobic foam was modified with POSS-OH, which achieved a contact angle of 140° and water absorption at the lowest level (11.2% after 24 h). This is due to the presence of non-polar side chains in the corners of silsesquioxane cages, which reduces the surface energy of the entire system.

## 4. Conclusions

RPUFs were successfully reinforced using POSS with hydroxyl and amino groups. The impact of POSSs on thermal properties, dynamic mechanical properties, physico-mechanical properties (compressive strength, three-point bending test, impact strength apparent density), foaming parameters and morphology of RPUFs was examined. The presented results indicate that the addition of AP-POSS, TS-POSS, and OH-POSS in the range of 0.5 wt.% influences the morphology of analyzed foams and consequently their further mechanical and thermal properties. It was noticed that RPUFs modified with AP-POSS are characterized by smaller and more regular polyurethane cells. This suggests better compatibility between PU foam matrix and AP-POSS compared with other fillers. This results in significant improvement of physico-mechanical properties and thermal stability of composites with AP-POSS. For example, compared to the RPUFs modified with OH-POSS and TS-POSS, composition with 0.5 wt.% of the AP-POSS showed greater compressive strength (351 kPa) and higher flexural strength (0.469 MPa). However, the highest hydrophobicity showed OH-PU foams, which were characterized by the greatest contact angle (140°) and less water uptake (11.2% after 24 h).

## Figures and Tables

**Figure 1 polymers-11-01092-f001:**
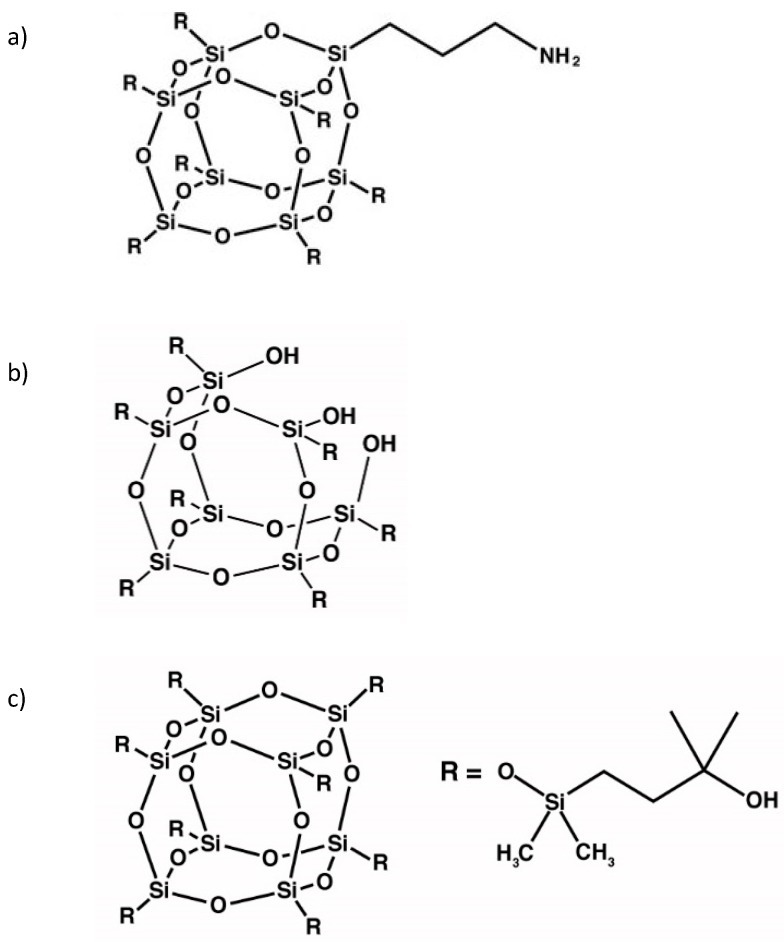
Chemical structure of (**a**) AP-POSS, (**b**) TS-POSS and (**c**) OH-POSS.

**Figure 2 polymers-11-01092-f002:**
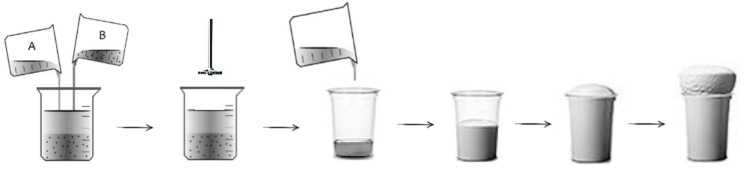
Schematic procedure of synthesis of rigid polyurethane foams (RPUFs).

**Figure 3 polymers-11-01092-f003:**
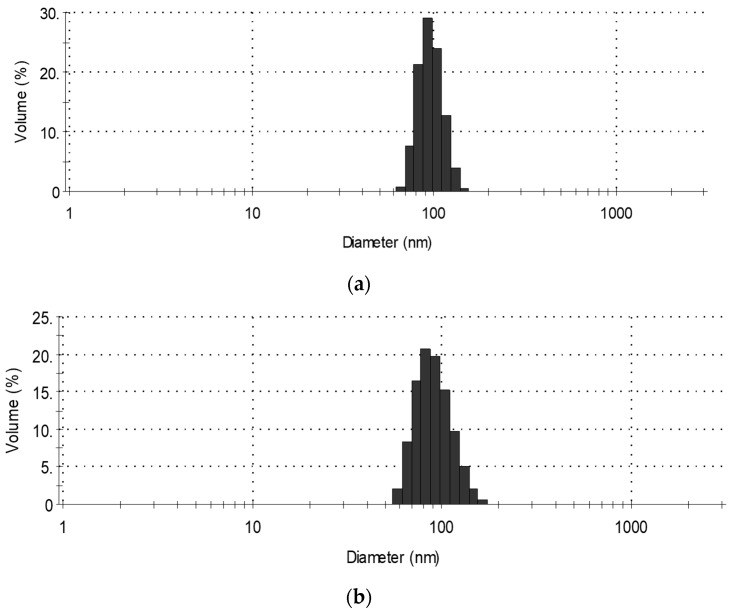
Distribution of particles size of (**a**) AP-POSS, (**b**) TS-POSS.

**Figure 4 polymers-11-01092-f004:**
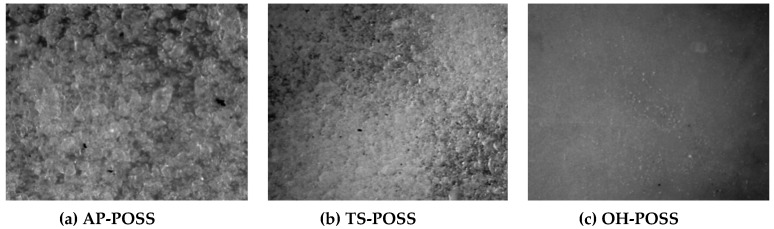
Polyol premixes with 0.5 wt.% of (**a**) AP-POSS, (**b**) TS-POSS and (**c**) OH-POSS.

**Figure 5 polymers-11-01092-f005:**
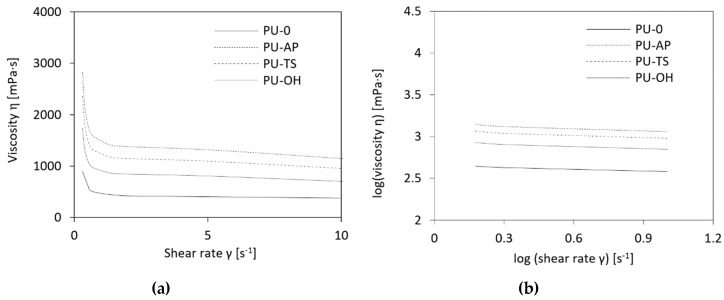
(**a**) Viscosity as a function of shear rate and (**b**) log-log plot of the viscosity vs. the shear rate for polyol premixes.

**Figure 6 polymers-11-01092-f006:**
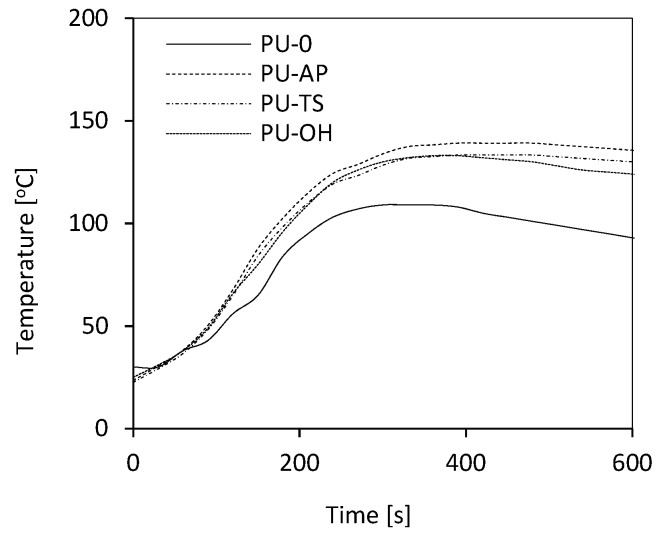
Temperature of reaction mixture in PU formulations.

**Figure 7 polymers-11-01092-f007:**
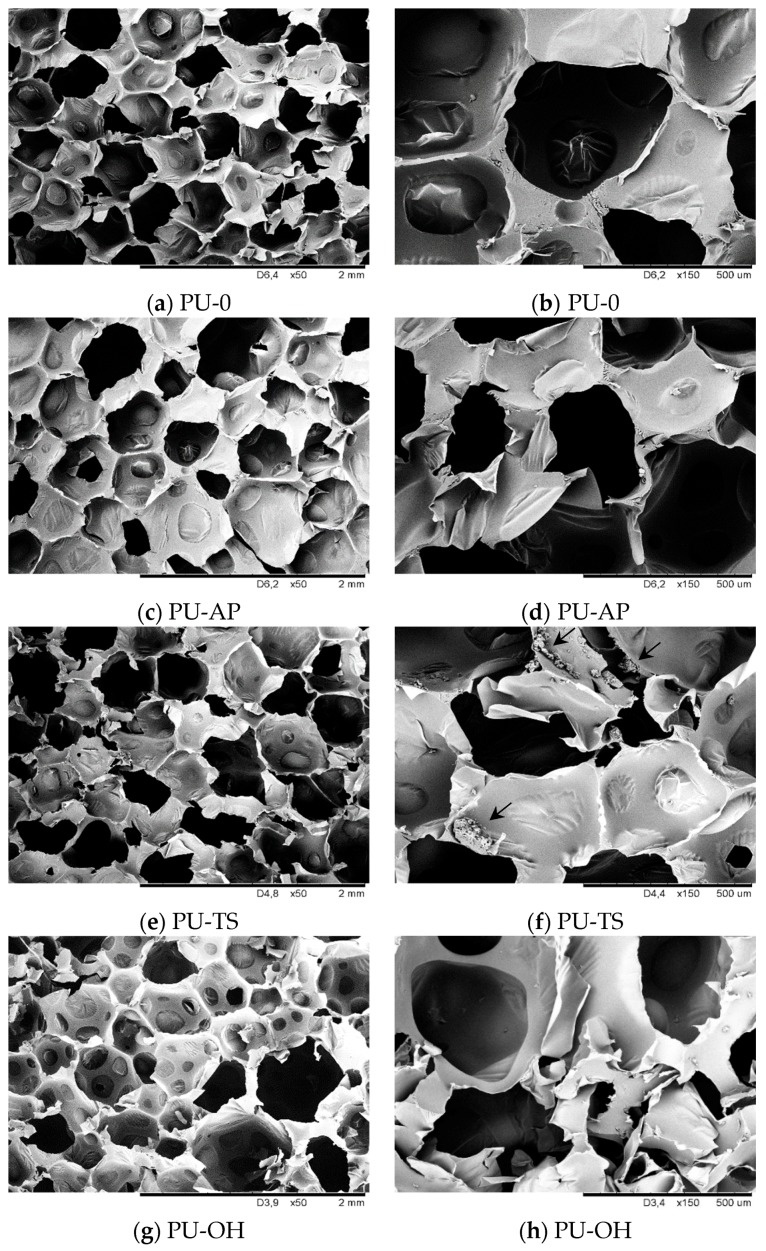
Morphology of (**a**,**b**) PU-0; (**c**,**d**) PU-AP, (**e**,**f**) PU-TS, (**g**,**h**) PU-OH observed at different magnification.

**Figure 8 polymers-11-01092-f008:**
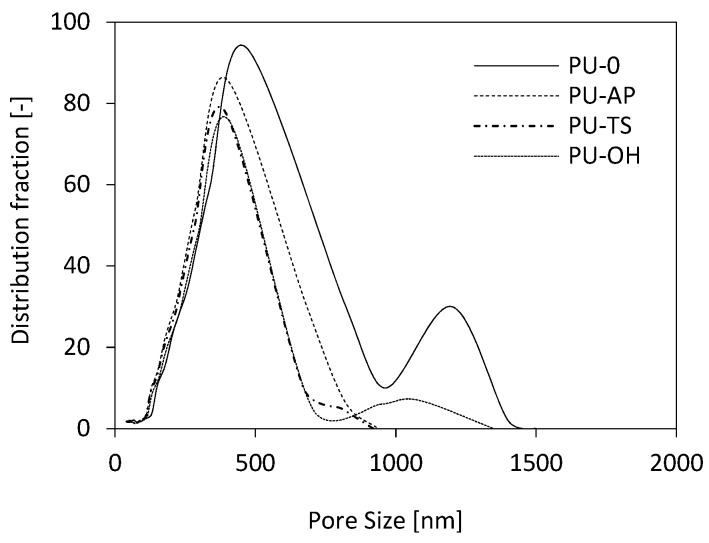
Cell size distributions of RPUFs.

**Figure 9 polymers-11-01092-f009:**
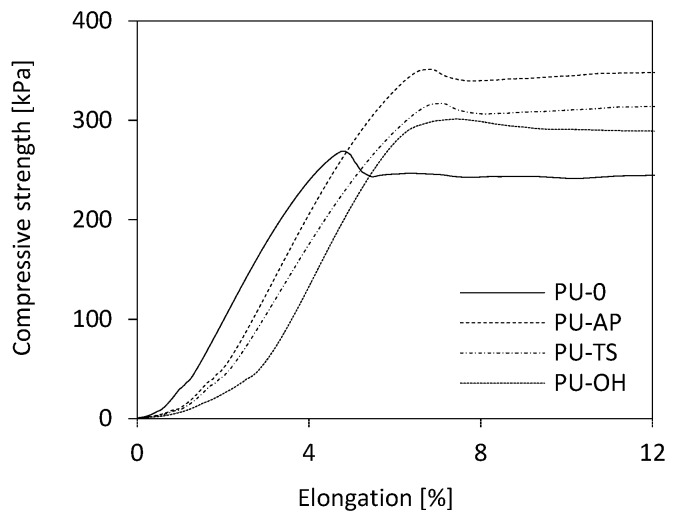
Compression behaviors of RPUFs measured parallel to the foam rise direction.

**Figure 10 polymers-11-01092-f010:**
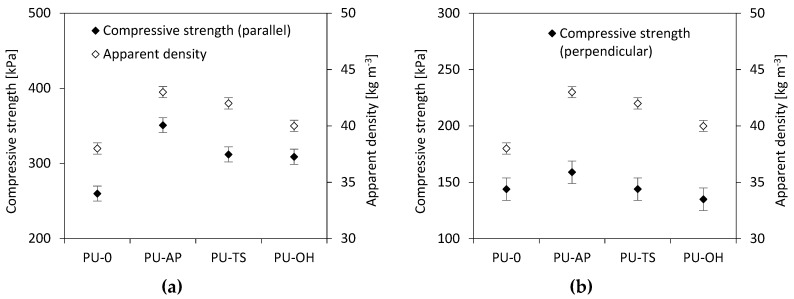
Effect of apparent density on compressive strength of RPUFs of RPUFs measured (**a**) parallel and (**b**) perpendicular to the foam rise direction.

**Figure 11 polymers-11-01092-f011:**
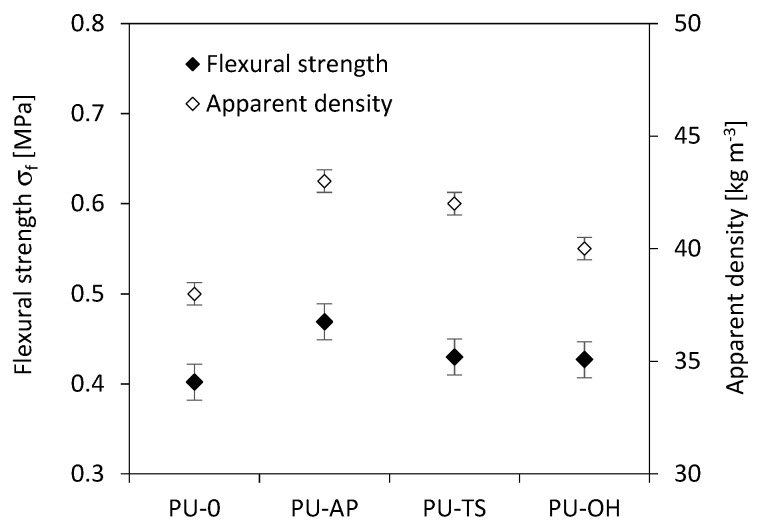
Effect of apparent density on flexural strength (*σ_f_*) of RPUFs.

**Figure 12 polymers-11-01092-f012:**
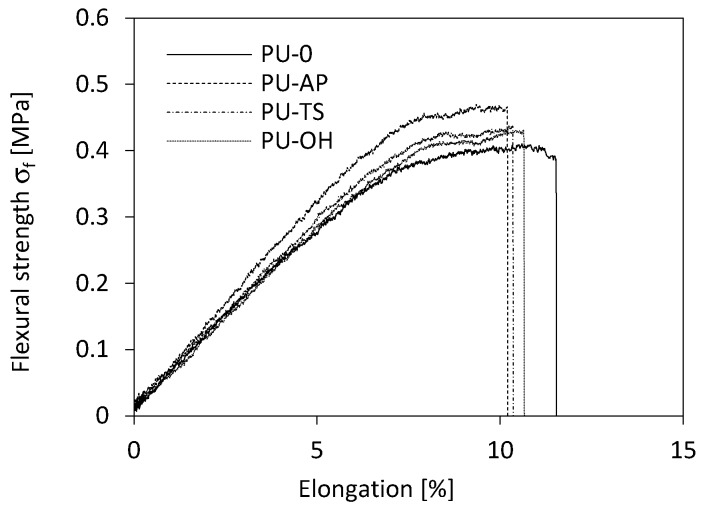
Flexural stress-elongation curves of RPUFs.

**Figure 13 polymers-11-01092-f013:**
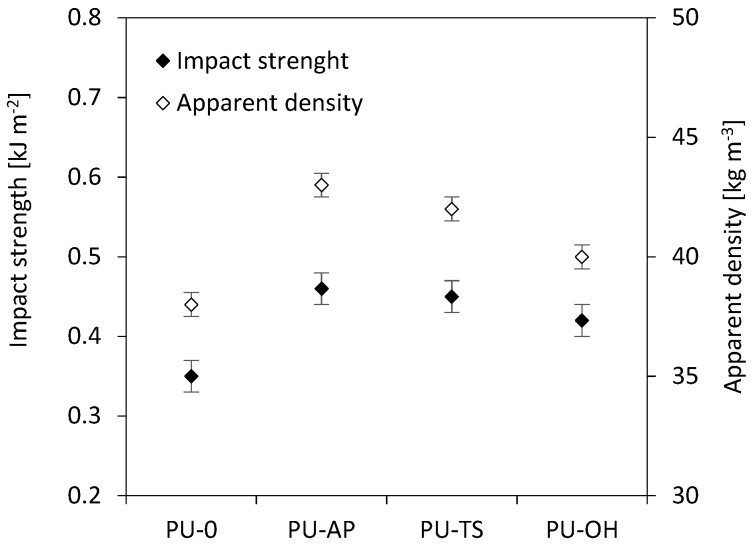
Effect of apparent density on the impact strength of RPUFs.

**Figure 14 polymers-11-01092-f014:**
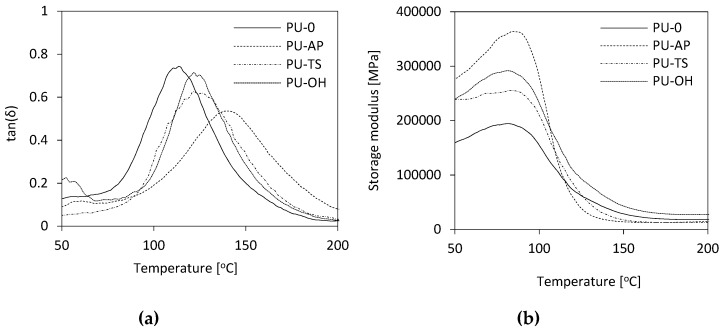
(**a**) tanδ and (**b**) storage modulus as a function of temperature plotted for RPUFs.

**Figure 15 polymers-11-01092-f015:**
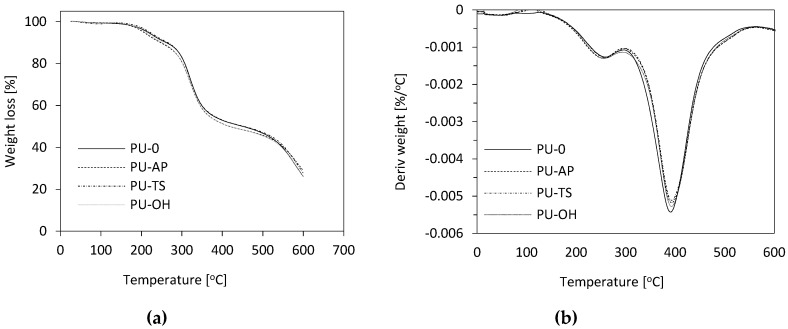
(**a**) TGA and (**b**) DTG curves for RPUFs modified with POSS.

**Figure 16 polymers-11-01092-f016:**
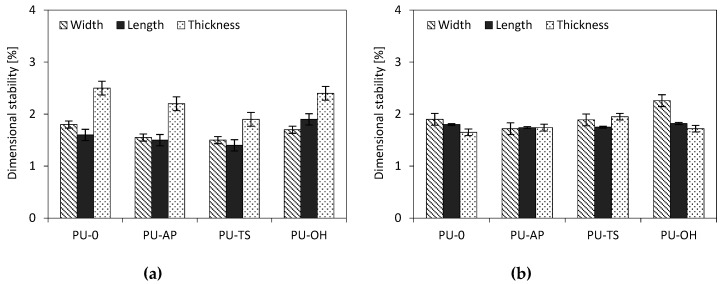
Dimensional stability of RPUFs modified with AP-POSS, TS-POSS and OH-POSS after exposure at 70 °C (**a**) and −20 °C (**b**).

**Figure 17 polymers-11-01092-f017:**
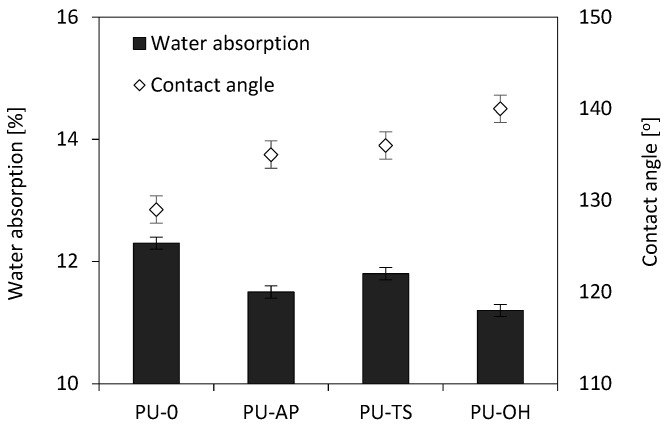
Effect of contact angle on water absorption of RPUFs modified with POSS.

**Figure 18 polymers-11-01092-f018:**
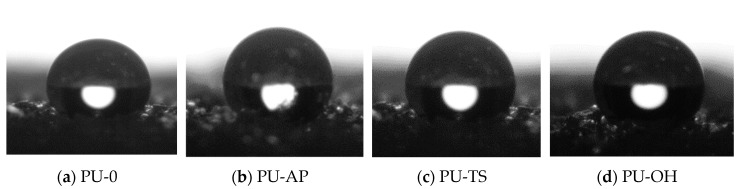
The contact angle of the surface of the (**a**) PU-0, (**b**) PU-AP, (**c**) PU-TS, (**d**) PU-OH.

**Table 1 polymers-11-01092-t001:** Dynamic viscosity and logarithmic plot of the fitting equations for polyol premixes.

Sample Code	Dynamic Viscosity *η* [mPa·s]			Fitting Equation	Power Law Index (n)	R^2^
	0.5 RPM	5 RPM	10 RPM			
PU-0	628	424	380	y = −0.058 + 0.335	0.335	0.982
PU-AP	2110	1313	1149	y = −0.060 + 0.318	0.298	0.978
PU-TS	1317	1098	957	y = −0.060 + 0.315	0.315	0.978
PU-OH	1149	805	702	y = −0.059 + 0.298	0.318	0.979

**Table 2 polymers-11-01092-t002:** Selected properties of RPUFs.

Sample Code	Temperature [°C]	Cream Time [s]	Extension Time [s]	Tack-free Time [s]	Cell Size [µm]	Wall Thickness [µm]	Apparent Density [kg m^−3^]
PU-0	110	43 ± 4	277 ± 10	341 ± 14	472 ± 10	62 ± 4	38
PU-AP	135	49 ± 2	512 ± 11	376 ± 12	390 ± 8	68 ± 2	43
PU-TS	136	47 ± 2	504 ± 8	370 ± 12	402 ± 6	66 ± 3	42
PU-OH	134	46 ± 2	428 ± 9	320 ± 10	410 ± 8	66 ± 2	40

**Table 3 polymers-11-01092-t003:** Mechanical properties of RPUFs.

Sample Code	Compressive Strength (Parallel) σ_10_ [kPa]	Compressive Strength (Perpendicular) σ_10_ [kPa]	Young Modulus [MPa]	Flexural Strength σ_f_ [MPa]	Elongation [%]	Impact Strength [kJ m^−2^]
PU-0	260	144	5	0.402	11.2	0.35
PU-AP	351	159	6.1	0.469	10.2	0.46
PU-TS	312	144	5.4	0.430	10.4	0.45
PU-OH	309	135	5.2	0.427	10.8	0.42

**Table 4 polymers-11-01092-t004:** The results of the thermogravimetric analysis of RPUFs.

Sample Code	*T_g_*	*T_5%_*	*T_10%_*	*T_50%_*	*T_70%_*	*Char Residue*
	[°C]	[°C]	[°C]	[°C]	[°C]	[%]
AP-POSS	-	267	280	342	531	21.6
TS-POSS	-	260	287	350	449	18.4
OH-POSS	-	n.d.	n.d.	n.d.	n.d.	n.d.
PU-0	112	220	265	454	591	27.9
PU-AP	137	205	245	418	595	29.0
PU-TS	127	216	261	451	586	28.7
PU-OH	121	210	251	449	585	28.6

**Table 5 polymers-11-01092-t005:** Dimensional stability and affinity for water of RPUFs.

Sample Code	Dimensional Stability (+70°C) [%]			Dimensional Stability (−20°C) [%]			Water Absorption [%]	Contact Angle [°]
	Width	Length	Thickness	Width	Length	Thickness		
PU-0	1.80	1.6	2.5	1.90	1.8	1.65	12.3	129
PU-AP	1.55	1.5	2.2	1.72	1.74	1.74	11.5	135
PU-TS	1.5	1.4	1.90	1.89	1.75	1.95	11.8	136
PU-OH	1.7	1.9	2.4	2.26	1.82	1.72	11.2	140

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
