# Peer review of "POSS Compounds as Modifiers for Rigid Polyurethane Foams (Composites)"

_polymers, 2019, doi:10.3390/polym11071092_

Round 1

Reviewer 1 Report

The author developed three types of functionalized POSS to modify RPUFs, and the processing parameters, morphology, mechanical properties, viscoelastic behavior, thermal property and applications of the obtained foams are discussed. The topic is within the scope of Polymers, and the results are consistent. Several points should be addressed before accepted:

1. Some recently published key developments on the PU foams should be mentioned and cited in the introduction part, e.g., Polymer, 2019, 170, 65-75.

2. Figure 3 is too small to see it. Please provide a clearer version in the revised manuscript.

3. Which type of POSS is more effective in this work? Is it also have the same reinforcing effect on various polymer systems?

4. The manuscript is well written, but I still suggest before it is considered to be accepted, all the sentences should be check carefully to make sure it can be read with pleasure.

Author Response

Dear Reviewer, thank you for the valuable analysis of the presented article. The authors will try their best to refer to all comments.

Reviewer 2 Report

This manuscript deals with the modification of rigid Polyurethane foams by using POSSs, thus obtaining hybrid composites, which were widely characterized. The text is quite well written and the Introduction quite well descriptive of the state of the art (probably can be enriched with some references, see attached .pdf). I suggest defining abbreviations first time are used, in both the abstract and the text, and then it is not necessary to repeat them. I suggest also homogenizing the use of verbs, by preferring past tense. Other suggestions are reported in the attached .pdf, such as the reducing of the axis in TG thermogram and the checking of some references that are affected by typos. Preparation and characterization of the composites were carried out correctly and the discussion is interesting and in agreement with the conclusions that are worthy of publication. Some minor changes are suggested (see attached .pdf) before acceptance for publication.

Author Response

(The authors gave the same response as above.)
